# Increased Incidence and Associated Risk Factors of Aspergillosis in Patients with Bronchiectasis

**DOI:** 10.3390/jpm11050422

**Published:** 2021-05-17

**Authors:** Bumhee Yang, Taehee Kim, Jiin Ryu, Hye Yun Park, Bin Hwangbo, Sun-Young Kong, Yong-Soo Kwon, Seung Jun Lee, Seung Won Ra, Yeon-Mok Oh, Jang Won Sohn, Kang Hyeon Choe, Hayoung Choi, Hyun Lee

**Affiliations:** 1Division of Pulmonary and Critical Care Medicine, Department of Internal Medicine, Chungbuk National University Hospital, College of Medicine, Chungbuk National University, Cheongju 28644, Korea; ybhworld0415@gmail.com (B.Y.); choekh@chungbuk.ac.kr (K.H.C.); 2Division of Pulmonary, Allergy, and Critical Care Medicine, Department of Internal Medicine, Hallym University Kangnam Sacred Heart Hospital, College of Medicine, Hallym University, Seoul 07441, Korea; sosom02@naver.com (T.K.); jwsohn@hanyang.ac.kr (J.W.S.); 3Biostatistical Consulting and Research Lab, Medical Research Collaborating Center, Hanyang University, Seoul 04763, Korea; ryu3209@hanyang.ac.kr; 4Division of Pulmonary and Critical Care Medicine, Department of Medicine, Samsung Medical Center, School of Medicine, Sungkyunkwan University, Seoul 06351, Korea; hyeyunpark@skku.edu; 5Division of Pulmonology, Center of Lung Cancer, Hospital, National Cancer Center, Goyang 10408, Korea; hbb@ncc.re.kr; 6Department of Laboratory Medicine, Hospital, National Cancer Center, Goyang 10408, Korea; ksy@ncc.re.kr; 7Department of Internal Medicine, Chonnam National University Hospital, Gwangju 61469, Korea; yskwon@chonnam.ac.kr; 8Division of Pulmonology and Allergy, Department of Internal Medicine, Gyeongsang National University Hospital, School of Medicine, Gyeongsang National University, Jinju 52727, Korea; juny2278@naver.com; 9Division of Pulmonary Medicine, Department of Internal Medicine, Ulsan University Hospital, College of Medicine, University of Ulsan, Ulsan 44033, Korea; skydocra@gmail.com; 10Department of Pulmonary and Critical Care Medicine, Asan Medical Center, College of Medicine, University of Ulsan, Seoul 05505, Korea; yeonmok.oh@gmail.com

**Keywords:** aspergillosis, bronchiectasis, epidemiology, risk factors

## Abstract

There are insufficient data regarding the relationship between non-cystic fibrosis bronchiectasis and incident aspergillosis. We performed a population-based, matched cohort study using data from the Korean National Health Insurance database between 2003 and 2013. The incidence of aspergillosis was 50/100,000 person-years in the bronchiectasis cohort and 11/100,000 person-years in the matched cohort (subdistribution hazard ratio, 4.53; 95% confidence interval (CI), 3.25–6.32). Among the bronchiectasis cohort, chronic obstructive pulmonary disease (adjusted HR, 1.95; 95% CI, 1.07–3.57), previous pulmonary tuberculosis (adjusted HR, 3.67; 95% CI, 2.03–6.64), and non-tuberculous mycobacterial pulmonary disease (adjusted HR, 11.25; 95% CI, 1.49–85.18) increased the risk of incident aspergillosis. The incidence of aspergillosis in patients with bronchiectasis was approximately 4.5-fold that in those without bronchiectasis. Comorbid pulmonary diseases—chronic obstructive pulmonary disease, previous pulmonary tuberculosis, and non-tuberculous mycobacterial pulmonary disease—significantly increased the risk of aspergillosis in patients with bronchiectasis. Our study indicates that close monitoring is warranted for aspergillosis in patients with bronchiectasis.

## 1. Introduction

Non-cystic fibrosis bronchiectasis (hereafter referred to as bronchiectasis) is a chronic lung disease characterised by permanent bronchial dilatation, variable mucociliary clearance, and recurrent infections [1,2]. The inherent dysfunctional mucociliary clearance leads to persistent bacterial infection, chronic inflammation of the bronchial tree, and progressive tissue destruction [3]. Therefore, patients with bronchiectasis are susceptible to many infections, such as bacterial and non-tuberculous mycobacterium (NTM) infections [4,5,6]. In addition to these bacteria, filamentous *Aspergillus* species, the most common fungi isolated from the sputum of patients with bronchiectasis, are also recognised as potential pathogens in bronchiectasis [3].

Of a number of manifestations of *Aspergillus* disease encountered in patients with bronchiectasis, the most common comorbid infectious conditions include pulmonary aspergilloma, chronic pulmonary aspergillosis (CPA), and invasive aspergillosis [7,8,9]. Considering that patients with bronchiectasis with these types of aspergillosis may benefit from treatments such as antifungal treatment, it is important to elucidate whether bronchiectasis increases the risk of aspergillosis and to identify risk factors related to the process. However, there is no concrete evidence of this issue.

The identification of risk factors predicting comorbid conditions in patients with bronchiectasis is important for establishing strategies to monitor and prevent these conditions. It would also facilitate early therapeutic intervention when aspergillosis develops in patients with bronchiectasis. However, little information is available regarding the risk factors for aspergillosis in patients with bronchiectasis, except for allergic bronchopulmonary aspergillosis (ABPA). Thus, elucidation of these risk factors is urgently needed.

In this study, we aimed to investigate whether patients with bronchiectasis are at a higher risk of aspergillosis than those without bronchiectasis and to identify risk factors for incident aspergillosis in patients with bronchiectasis.

## 2. Materials and Methods

### 2.1. Data Source

We used the National Health Insurance Service-National Sample Cohort (NHIS-NSC), a population-based retrospective cohort that included a 2.2% representative sample of all Korean citizens. The NHIS-NSC database includes health data regarding (1) major and minor diagnoses in the form of the 10th revision of the International Statistical Classification of Diseases and Related Health Problems (ICD-10) codes, and (2) procedures and drug prescriptions in outpatient clinics, emergency departments, and hospitalisations, in addition to age, sex, and type of insurance [10].

### 2.2. Study Population

As shown in Figure 1, between 1 January 2003 and 31 December 2012, the database included data for 861,808 adult patients aged ≥ 20 years. After exclusion of patients with human immunodeficiency virus infection (*n* = 627) and cystic fibrosis (*n* = 88), the sample included 17,679 patients with bronchiectasis and 843,414 without bronchiectasis. Among the 17,679 patients with bronchiectasis, we excluded those diagnosed with aspergillosis within 1 year after enrolment (*n* = 90) and those who died within 1 year after enrolment (*n* = 683). Among 843,414 patients without bronchiectasis, we excluded those diagnosed with aspergillosis within 1 year after enrolment in the cohort (*n* = 56) and those who died within 1 year after enrolment in the cohort (*n* = 4663). To establish a matched cohort, we performed a 1:4 matching for age and sex for each bronchiectasis patient and controls without bronchiectasis. The aspergillosis-naive bronchiectasis cohort (*n* = 16,906) and the aspergillosis-naive matched cohort (*n* = 67,624) were followed until 31 December 2013 to compare the incidence of aspergillosis between them. As the NHIS-NSC database contains all claim data of participants during the study period (2002–2013), only deceased or emigrated participants were excluded from the dataset [10]. None of the patients had a follow-up duration of less than 1 year.

The study protocol was approved by the Institutional Review Board of Chungbuk National University Hospital (IRB application no. 2020-12-027). The requirement for informed consent from the participants was waived because the NHIS database was constructed after anonymization.

### 2.3. Definitions

As exposure of this study, adult bronchiectasis was defined by the following criteria: (1) age ≥20 years and (2) at least one claim under ICD-10 diagnosis code J47. Patients with cystic fibrosis (E84) and human immunodeficiency virus infection (B20–B24) were excluded from this study.

As an outcome of this study, aspergillosis was defined by the presence of a primary or secondary diagnostic code associated with aspergillosis (B44). Patients with tonsillar aspergillosis (B44.2) or ABPA (B44.81) were excluded.

Bronchiectasis-associated comorbidities were also defined using the following ICD-10 diagnosis codes: chronic obstructive pulmonary disease (COPD) (J42–J44, except J43.0 (unilateral emphysema)), asthma (J45–J46), pulmonary tuberculosis (TB) (A15–A19), non-tuberculous mycobacterial pulmonary disease (NTM-PD) (A31 and claims data for acid-fast bacilli smear or mycobacterial culture), diabetes mellitus (E10–E14), rheumatologic disease (M05–M06, M315, M32–M34, M351, M353, and M360), and malignancy (C00–C97), including lung cancer (C34) [11,12].

Regarding corticosteroid use, the study population was categorised into three groups according to the use of inhaled corticosteroids (ICS) (no ICS use, ICS use for less than 1 year, and ICS use for at least 1 year) and systemic corticosteroids (no systemic corticosteroid use, <10 mg/day prednisolone equivalent dose, and >10 mg/day prednisolone equivalent dose) [13].

### 2.4. Statistical Analysis

The McNemar test was used to compare baseline characteristics between the bronchiectasis cohort and the matched cohort. To construct the aspergillosis-naive bronchiectasis cohort, we excluded patients diagnosed with aspergillosis within 1 year after enrolment in the bronchiectasis cohort. Given that 1 year might not be enough to construct an aspergillosis-naive bronchiectasis cohort, we performed a sensitivity analysis with different exclusion criteria using 2- and 3-year washout periods.

We compared the incidence rate (per 100,000 person-years (PY)) of aspergillosis between the two cohorts using the normal approximation test for binominals. The cumulative incidence of aspergillosis was compared between the bronchiectasis cohort and the matched cohort using the cumulative incidence curve and Gray’s test. To evaluate the impact of bronchiectasis on incident aspergillosis in the bronchiectasis cohort relative to the matched cohort, a proportional subdistribution hazards model for aspergillosis with death as a competing event was used.

To analyse the risk factors associated with aspergillosis, we performed Cox proportional hazards regression models. In multivariable models, variables with *p* < 0.05 in univariable analysis were included: age, asthma, COPD, previous pulmonary TB, NTM-PD, use of ICS, and systemic corticosteroid use. All tests were two-tailed, and *p*-values < 0.05 were considered statistically significant. All statistical analyses were performed using SAS^®^ software (version 9.4; SAS Institute, Cary, NC, USA).

## 3. Results

### 3.1. Baseline Characteristics

As shown in Table 1, there were no significant intergroup differences between the bronchiectasis and matched cohorts in age and sex. The proportion of patients who received medical aid was higher in the bronchiectasis cohort than in the matched cohort (2.9% vs. 2.3%, *p* < 0.001). Regarding pulmonary comorbidities, the proportions of asthma (34.4% vs. 8.9%), COPD (27.7% vs. 5.2%), previous pulmonary TB (12.0% vs. 2.6%), and NTM-PD (0.1% vs. <0.1%) were higher in the bronchiectasis cohort than in the matched cohort (*p* < 0.001 for all). Regarding extrapulmonary comorbidities, diabetes mellitus (20.4% vs. 15.8%), rheumatologic diseases (6.1% vs. 3.4%), lung cancer (3.8% vs. 0.3%), and other cancers (3.7% vs. 2.2%) were significantly more frequent in the bronchiectasis cohort than in the matched cohort (*p* < 0.001 for all).

### 3.2. Incidence Rate and Risk of Aspergillosis in the Bronchiectasis Cohort Versus the Matched Cohort

During the median follow-up duration of 6.1 (interquartile range, 3.4–8.7) years, the age- and sex-adjusted incidence of aspergillosis was 50.2/100,000 PY in the bronchiectasis cohort and 10.9/100,000 PY in the matched cohort (subdistribution hazard ratio (sdHR), 4.53; 95% confidence interval (CI), 3.25–6.32). The risk of aspergillosis was higher in patients with bronchiectasis aged <60 years (sdHR, 4.91; 95% CI, 3.07–7.84) than in those aged ≥60 years (sdHR, 4.17; 95% CI, 2.60–6.71) (*p* for interaction < 0.01). The risk for incident aspergillosis was higher among females in the bronchiectasis cohort (sdHR, 5.33; 95% CI, 3.21–8.87) than among males in the bronchiectasis cohort (sdHR, 4.00; 95% CI, 2.57–6.23) (*p* for interaction < 0.01) (Table 2). Sensitivity analyses also revealed comparable HRs for incident aspergillosis in patients with bronchiectasis relative to those without bronchiectasis: 2-year (sdHR, 4.94; 95% CI, 3.16–7.71) and 3-year (sdHR, 4.94; 95% CI, 3.02–8.08) washout periods, respectively (Appendix A).

As shown in Figure 2, the cumulative incidence of aspergillosis was significantly higher in the bronchiectasis cohort than in the matched cohort (Gray’s test, *p* < 0.001). Subgroup analyses also revealed that the cumulative incidence of aspergillosis was significantly higher in the bronchiectasis cohort than in the matched cohort, regardless of age group or sex (Gray’s test, *p* < 0.001 for all subgroups, Appendix A). Detailed information on the diagnosis of aspergillosis in the bronchiectasis cohort and the matched cohort are summarised in Appendix A.

### 3.3. Risk Factors of Aspergillosis in the Bronchiectasis Cohort

Table 3 summarises the risk factors for aspergillosis in the bronchiectasis cohort. Although age was significantly associated with an increased risk of aspergillosis in the bronchiectasis cohort in univariable analysis, this association was not statistically significant in the multivariable analysis. Multivariable analysis revealed that pulmonary comorbidities—COPD (adjusted hazard ratio (HR), 1.95; 95% CI, 1.07–3.57), previous pulmonary TB (adjusted HR, 3.67; 95% CI, 2.03–6.64), and NTM-PD (adjusted HR, 11.25; 95% CI, 1.49–85.18)—significantly increased the risk of aspergillosis.

## 4. Discussion

In the present study, we compared the incidence of aspergillosis between patients with bronchiectasis and age- and sex-matched patients without bronchiectasis. Our results show that the risk of aspergillosis in patients with bronchiectasis was approximately 4.5-fold higher than that in those without bronchiectasis. Compared to patients without bronchiectasis, the risk of aspergillosis in those with bronchiectasis was especially higher in those aged < 60 years and women. Regarding risk factors, COPD, previous pulmonary TB, and NTM-PD significantly increased the risk of aspergillosis in patients with bronchiectasis.

To the best of our knowledge, this is the first study to assess the incidence and risk factors of incident aspergillosis in patients with bronchiectasis using a nationally representative cohort. *Aspergillus* infections, including *Aspergillus* bronchitis and CPA, are not rare in patients with bronchiectasis, and it has been suggested that they either cause bronchiectasis or complicate existing bronchiectasis [9,14]. However, the burden of aspergillosis among patients with bronchiectasis has not been well studied as the focus of previous studies was mainly limited to ABPA and did not include comorbid aspergillosis [11,15,16,17]. Data regarding the relationship between bronchiectasis and *Aspergillus* infection were considered in studies that evaluated bronchiectasis as a comorbid pulmonary disease in patients with aspergillosis. In these studies, the proportion of patients with bronchiectasis was 38% among patients with CPA and 14.5% among patients with pulmonary aspergillosis [18,19]. However, because of their cross-sectional design, these previous studies could not clarify whether there is a cause-and-effect link between these two conditions. The present study has a substantial advantage in terms of causal inference in that approximately 0.3% of bronchiectasis cases without aspergillosis develop aspergillosis during a follow-up duration of about six years.

In this study, the risk of aspergillosis in patients with bronchiectasis compared to those without bronchiectasis was particularly higher among younger patients (less than 60 years) and females. In younger patients, the prevalence of comorbid conditions (e.g., diabetes mellitus and malignancy), which increases susceptibility to aspergillosis, was lower than in the elderly population. Thus, the effect of bronchiectasis on the development of aspergillosis might be more influential in younger patients than in older patients. In line with our findings, a previous study showed that the positive rates of *A. fumigatus*-specific immunoglobulin G Ab were higher in young and female patients among patients with TB [12]. The present study and a previous study suggest that the risk of aspergillosis is higher in women than in men when structural bronchial abnormalities, such as bronchiectasis and TB sequelae, are present. Accordingly, our results may have important clinical implications, indicating that more attention is needed regarding surveillance for aspergillosis development among younger or female patients with bronchiectasis. However, except for the current study, no studies have evaluated the susceptibility to aspergillosis by age or sex in patients with bronchiectasis compared to those without bronchiectasis. Thus, further studies are needed to confirm our findings.

We evaluated the risk factors of aspergillosis in patients with bronchiectasis, and our results provided valuable information for clinicians assessing the risk of aspergillosis in patients with bronchiectasis with respect to comorbid conditions, including COPD, TB, and NTM-PD. COPD is a chronic inflammatory disease of the airways [20], and airway colonisation by *Aspergillus* species is a common feature of COPD [21]. Thus, COPD patients have been recognised to be at an increased risk of pulmonary aspergillosis [22]. Additionally, some previous studies reported that COPD is an underlying pulmonary comorbid condition in 42–67% of patients with pulmonary aspergillosis [23,24]. In agreement with previous studies, we found that COPD increased the risk of aspergillosis 2-fold in patients with bronchiectasis.

A significant association between CPA development and post-TB patients was shown in a cohort study in the 1960s; among 544 TB patients with persistent cavities after treatment, 25% had positive Aspergillus precipitins in blood samples, and 14% had aspergillomas in radiological images [25]. CPA is one of the most common infectious sequelae in post-TB patients, with an estimated prevalence of 1.2 million cases annually worldwide, indicating a significant, newly emerging global disease burden [26]. Sehgal et al. also showed a significant occurrence of Aspergillus sensitisation and chronic infection in patients with bronchiectasis, especially in those with post-TB bronchiectasis [27]. However, the associations of post-TB sequelae, bronchiectasis, and pulmonary aspergillosis are complicated as these conditions interact. Conflicting results have also been reported in a previous study. A study performed in Uganda found that approximately 5% of post-TB patients were diagnosed with CPA during a 2-year follow-up period. Although bronchiectasis was present in 86% of patients, it was not associated with an increased risk of CPA [28]. In our study, we observed that previous TB history increased the risk of aspergillosis 4-fold in patients with bronchiectasis. After TB treatment, residual lung damage in the form of bronchiectasis may lead to chronic Aspergillus colonisation and sensitisation, followed by aspergillosis infection. Hence, clinicians should be aware of the risk of aspergillosis when managing patients with bronchiectasis, especially those with a history of TB. Such awareness will allow clinicians to make early diagnoses in patients with aspergillosis who benefit from antifungal treatment.

In previous studies, strong associations were observed between Aspergillus infection and NTM-PD. A cross-sectional study reported that among 126 CPA patients, 14.9% had NTM-PD, making it one of the most common underlying lung diseases [29]. Another case-control study also showed that positivity for Aspergillus serology in patients with bronchiectasis with NTM-PD was 5-fold that in those without NTM-PD [14]. In agreement with these results, we observed that NTM-PD was associated with an increased risk of aspergillosis in patients with bronchiectasis. However, only 1 of 19 patients with NTM-PD developed aspergillosis. As the number of patients with NTM-PD was small, our results should be interpreted with caution, and future studies with larger sample sizes are warranted to unveil the link between NTM-PD and increased risk of aspergillosis in patients with bronchiectasis.

ICS use has been regarded as a major risk factor for Aspergillus infection in patients with cystic fibrosis [30,31,32]. In a recent study, the use of ICS in patients with cystic fibrosis was associated with the isolation of Aspergillus fumigatus in sputum [33]. It has not been clear until now whether the use of ICS increases the risk of aspergillosis in patients with non-cystic fibrosis bronchiectasis. In this study, the use of ICS was associated with an increased risk of incident aspergillosis in univariable analysis; however, the association disappeared after adjustment for covariates in the multivariable analysis. Likewise, the small number of aspergillosis cases might have been related to the statistical insignificance of the use of systemic corticosteroids in patients with bronchiectasis. Thus, future studies are needed to clarify the relationship between corticosteroids use and the development of aspergillosis in patients with bronchiectasis.

In this study, it was necessary to exclude bronchiectasis cases in which aspergillosis was diagnosed within one year after the diagnosis of bronchiectasis (approximately 0.5% of patients with bronchiectasis) to establish the aspergillosis-naive bronchiectasis cohort. Nevertheless, this finding indicates that some patients with bronchiectasis had coexisting aspergillosis at the time of cohort enrolment. Thus, it is important to develop appropriate clinical strategies to diagnose aspergillosis in patients with newly diagnosed bronchiectasis. However, there is no consensus on this issue. Only the ABPA test for etiologic evaluation has been mentioned in the recently revised British Thoracic Society guidelines for bronchiectasis [34]. Therefore, further studies regarding the evaluation and management of aspergillosis when bronchiectasis is diagnosed are urgently required.

There are several limitations of this study, mostly due to the limitations of the NHIS data related to ICD-10 diagnosis codes. First, we were unable to obtain clinical, radiological, or microbiological information. Second, defining aspergillosis based on ICD-10 codes may underestimate the actual cases of aspergillosis. Third, the incidence of aspergillosis was relatively small, which might have led to a lack of statistical significance in the potential risk factors.

## 5. Conclusions

The incidence of aspergillosis in patients with bronchiectasis was approximately 5-fold that in patients without bronchiectasis. The risk was higher in the female and younger (<60 years) patients. The risk factors for incident aspergillosis in bronchiectasis were comorbid pulmonary diseases, such as COPD, previous pulmonary TB, and NTM-PD. When managing patients with bronchiectasis, clinicians should consider paying more attention to those with comorbid pulmonary diseases in order to diagnose aspergillosis early and plan effective treatment.

## Figures and Tables

**Figure 1 jpm-11-00422-f001:**
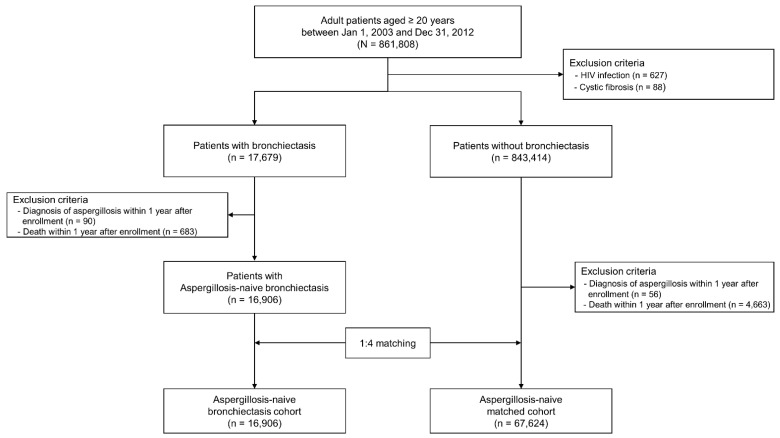
Flow chart of the study. HIV, human immunodeficiency virus.

**Figure 2 jpm-11-00422-f002:**
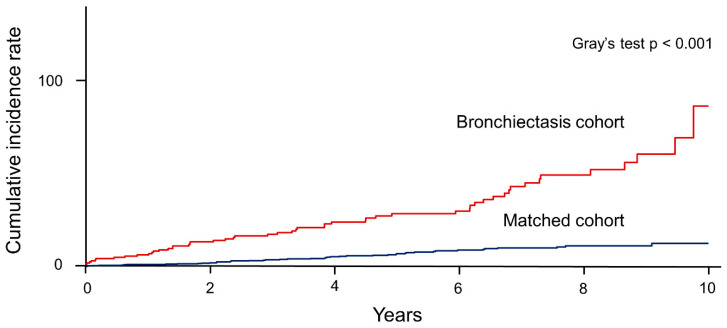
Cumulative incidence of aspergillosis (/100,000 person-years) in the bronchiectasis and matched cohorts.

**Table 1 jpm-11-00422-t001:** Baseline characteristics of patients.

	Total (*N* = 84,530)	Bronchiectasis Cohort (*n* = 16,906)	Matched Cohort (*n* = 67,624)	*p*-Value
Age, years				>0.999
20–29	3860 (4.6)	772 (4.6)	3088 (4.6)	
30–39	7840 (9.3)	1568 (9.3)	6272 (9.3)	
40–49	13,530 (16.0)	2706 (16.0)	10,824 (16.0)	
50–59	19,000 (22.5)	3800 (22.5)	15,200 (22.5)	
60–69	20,560 (24.3)	4112 (24.3)	16,448 (24.3)	
≥70	19,740 (23.3)	3948 (23.3)	15,792 (23.3)	
Sex				>0.999
Male	39,930 (47.2)	7986 (47.2)	31,944 (47.2)	
Female	44,600 (52.8)	8920 (52.8)	35,680 (52.8)	
Type of insurance				<0.001
Self-employed health insurance	32,097 (38.0)	6426 (38.0)	25,671 (38.0)	
Employee health insurance	50,384 (59.6)	9993 (59.1)	40,391 (59.7)	
Medical aid	2049 (2.4)	487 (2.9)	1562 (2.3)	
Pulmonary comorbidities				
Asthma	11,831 (14.0)	5819 (34.4)	6012 (8.9)	<0.001
COPD	8205 (9.7)	4679 (27.7)	3526 (5.2)	<0.001
Previous pulmonary tuberculosis	3776 (4.5)	2035 (12.0)	1741 (2.6)	<0.001
NTM pulmonary disease	21 (0.0)	19 (0.1)	2 (<0.1)	<0.001
Extra-pulmonary comorbidities				
Diabetes mellitus	14,134 (16.7)	3442 (20.4)	10,692 (15.8)	<0.001
Rheumatologic disease	3346 (4.0)	1022 (6.1)	2324 (3.4)	<0.001
Malignancy				
Lung cancer	843 (1.0)	639 (3.8)	204 (0.3)	<0.001
Other cancers	2119 (2.5)	632 (3.7)	1487 (2.2)	<0.001

Data are presented as number (percentage). Abbreviations: COPD, chronic obstructive lung disease; NTM, non-tuberculous mycobacteria.

**Table 2 jpm-11-00422-t002:** Sex- and age-specific incidence rates (/100,000 person-years) and subdistribution hazard ratios of aspergillosis in the bronchiectasis cohort relative to the matched cohort.

	Total (*N* = 84,675)				Male (*n* = 40,015)				Female (*n* = 44,660)			
	Case	IR	sdHR	95% CI	Case	IR	sdHR	95% CI	Case	IR	sdHR	95% CI
Overall												
Matched	45	10.9	Ref		27	14.2	Ref		18	8.1	Ref	
Bronchiectasis	51	50.2	4.53	3.25–6.32	27	57.9	4.00	2.57–6.23	24	43.7	5.33	3.21–8.87
Age group												
<60 years												
Matched	22	9.5	Ref		16	15.0	Ref		6	4.8	Ref	
Bronchiectasis	27	46.8	4.91	3.07–7.84	13	49.1	3.25	1.77–5.97	14	44.9	9.33	4.11–21.22
≥60 years												
Matched	23	12.8	Ref		11	13.1	Ref		12	12.5	Ref	
Bronchiectasis	24	54.7	4.17	2.60–6.71	14	69.3	5.09	2.64–9.82	10	42.2	3.33	1.66–6.69

Data are presented as risk ratios (95% confidence intervals). Matched and bronchiectasis denote the matched cohort and the bronchiectasis cohort, respectively. Abbreviations: IR, incidence rate; sdHR, subdistribution hazard ratio; CI, confidence interval; Ref, reference.

**Table 3 jpm-11-00422-t003:** Risk factors for aspergillosis in patients with bronchiectasis.

	Numbers at Risk (*N* = 16,906)	Aspergillosis (*n* = 51)	Univariable Analysis		Multivariable Analysis	
			HR	95% CI	Adjusted HR	95% CI
Age						
≤39	2340 (13.8)	2/2340 (0.1)	Ref	Ref	Ref	Ref
40–49	2706 (16.0)	11/2706 (0.4)	4.91	1.09–22.16	4.43	0.98–20.01
50–59	3800 (22.5)	14/3800 (0.4)	5.30	1.20–23.34	4.01	0.90–17.81
60–69	4112 (24.3)	13/4112 (0.3)	4.57	1.03–20.27	2.77	0.61–12.52
≥70	3948 (23.4)	11/3948 (0.3)	5.32	1.17–24.06	3.16	0.68–14.66
Sex						
Female	7986 (47.2)	27/7986 (0.3)	Ref	Ref		
Male	8920 (52.8)	24/8920 (0.3)	1.35	0.78–2.34		
Type of insurance						
Self-employed health insurance	6426 (38.0)	19/6426 (0.3)	Ref	Ref		
Employee health insurance	9993 (59.1)	31/9993 (0.3)	1.16	0.66–2.06		
Medical aid	487 (2.9)	1/487 (0.2)	1.73	0.23–13.02		
Comorbidities						
COPD	4679 (27.7)	27/4679 (0.6)	3.09	1.78–5.35	1.95	1.07–3.57
Asthma	5819 (34.4)	28/5819 (0.5)	2.27	1.31–3.93	1.27	0.68–2.37
Previous pulmonary tuberculosis	2035 (12.0)	19/2305 (0.8)	4.83	2.74–8.53	3.67	2.03–6.64
NTM pulmonary disease	19 (0.1)	1/19 (5.3)	28.73	3.96–208.41	11.25	1.49–85.18
Diabetes mellitus	3442 (20.4)	11/3442 (0.3)	1.32	0.68–2.58		
Rheumatologic disease	1022 (6.1)	3/1022 (0.3)	0.93	0.29–2.99		
Lung cancer	639 (3.8)	3/639 (0.5)	1.65	0.52–5.31		
Medication						
Use of ICS						
No use	13,149 (77.8)	27/13,149 (0.2)	Ref	Ref	Ref	Ref
<1 year	2540 (15.0)	14/2540 (0.6)	2.50	1.31–4.77	1.83	0.92–3.63
≥1 year	1217 (7.2)	10/1217 (0.8)	3.39	1.64–7.01	1.69	0.74–3.89
Systemic corticosteroids *						
No use	3602 (21.3)	5/3602 (0.1)	Ref	Ref	Ref	Ref
<10 mg/day	12,764 (75.5)	41/12,764 (0.3)	1.43	0.56–3.65	1.29	0.50–3.31
≥10 mg/day or more	540 (3.2)	5/540 (0.9)	3.60	1.04–12.50	2.15	0.59–7.86

Data are presented as number (percentage), ratio (percentage), or hazard ratios (95% CIs). * Systemic corticosteroids are presented as a prednisolone-equivalent dose. Abbreviations: COPD, chronic obstructive pulmonary disease; NTM, non-tuberculous mycobacteria; ICS, inhaled corticosteroids; HR, hazard ratio; CI, confidence interval; Ref, reference.

## Data Availability

The data presented in this study are available upon reasonable request from the corresponding author.

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
