# Peer review of "Increased Incidence and Associated Risk Factors of Aspergillosis in Patients with Bronchiectasis"

_jpm, 2021, doi:10.3390/jpm11050422_

Round 1
Reviewer 1 Report
This is a retrospective study that indicates close monitoring is needed for the occurrence of aspergillosis in patients with bronchiectasis.
It is well known that aspergillosis are more frequent in patients with bronchiectasis but this is a good paper to show this knowledge based in a national cohort study. I agree there are several limitations of this study, mostly due to limitations of the NHIS data related to ICD-10 diagnosis codes.
Some changes in English editing are required and I recommend to avoid figures 2b,2c,2d,2e to simplify the publication.
Author Response
Subject: Journal of Personalized Medicine - Manuscript Decision
Ref: Submission ID jpm-1180774
Dear Editor, Journal of Personalized Medicine
Thank you for your letter dated April 26, 2021. We appreciate the constructive review of our manuscript, entitled “Increased incidence and associated risk factors of aspergillosis in
patients with bronchiectasis” (Submission ID jpm-1180774), provided by reviewers. We are submitting a revised manuscript and a detailed point-by-point response is included below.
We look forward to responding to any additional comments or questions you may have concerning this paper and hope you find the revised manuscript acceptable for publication in Journal of Personalized Medicine.
Sincerely,
Hyun Lee, MD, on behalf of all authors.
Associate Professor
Division of Pulmonary Medicine and Allergy, Department of Internal Medicine, Hanyang University College of Medicine, 222-1, Wangsimni-ro, Seongdong-gu, Seoul 04763, Korea. E-mail: namuhanayeyo@hanyang.ac.kr
Responses to Comments
## Response to Reviewer 1’s comments
General comments
This is a retrospective study that indicates close monitoring is needed for the occurrence of aspergillosis in patients with bronchiectasis. It is well known that aspergillosis are more frequent in patients with bronchiectasis but this is a good paper to show this knowledge based in a national cohort study. I agree there are several limitations of this study, mostly due to limitations of the NHIS data related to ICD-10 diagnosis codes.
Response. Thank you for your positive comment. Regarding the concerns raised by the reviewers, we have provided point-by-point responses below.
Specific comments
Comment 1 (C1). Some changes in English editing are required and I recommend to avoid figures 2b,2c,2d,2e to simplify the publication.
Response 1-1 (R1-1). As recommended, the revised manuscript has been proofread by a professional editor who is a native English speaker.
R1-2. Thank you for your comment. As recommended, we have now reported the cumulative incidence rate of the overall population (Figure 2a in the original manuscript), as shown in Figure 2 in the revised manuscript. The cumulative incidence rates of subgroups (figure 2b, 2c, 2d, and 2e) are shown in Supplementary Figure S1 in the revised manuscript.
## Response to Reviewer 2’s comments
Specific comments
C1. Was there a minimum duration of follow-up for this cohort? If not, were patients who had follow-up duration < 1 year being excluded from this study?
R1. As the NHIS-NSC database contains all claims data of participants during the study period (2002–2013), only deceased or emigrated patients were excluded from the dataset [1]. None of the patients had a follow-up duration of less than 1 year. To address your concern, we have clarified this in the Methods section of the revised manuscript (page 4, lines 124–127).
Reference
- Lee, J.; Lee, J.S.; Park, S.H.; Shin, S.A.; Kim, K. Cohort Profile: The National Health Insurance Service-National Sample Cohort (NHIS-NSC), South Korea. Int J Epidemiol 2017, 46, e15, doi:10.1093/ije/dyv319.
“As the NHIS-NSC database contains all claim data of participants during the study period (2002–2013), only deceased or emigrated participants were excluded from the dataset. None of the patients had a follow-up duration of less than 1 year.”
C2. Why was diabetes mellitus not included in the Cox proportional hazards regression models?
R2. Thank you for pointing out that we did not fully acknowledge this in our original manuscript. As recommended, we evaluated whether diabetes mellitus was associated with an increased risk of aspergillosis. In the univariable Cox proportional hazards regression model, diabetes mellitus did not significantly increase the risk of aspergillosis (unadjusted HR, 1.32; 95% CI, 0.68–2.58). Thus, we did not include diabetes mellitus in the adjusted model. We have modified Table 3 in the revised manuscript accordingly (page 10).
C3. The statistical analyses for sensitivity analyses based on different criteria should be provided in the Methods section, with a summary of results presented in the Results section.
R3. Thank you for your comment. As recommended, we have provided statistical analyses for sensitivity analyses in the Methods section and summary in the Results section in the revised manuscript (page 5, lines 159–164 and page 7, lines 210–213).
page 5, lines 159–164
“To construct the aspergillosis-naive bronchiectasis cohort, we excluded patients diagnosed with aspergillosis within 1 year after enrolment in the bronchiectasis cohort. Given that 1 year might not be enough to construct an aspergillosis-naive bronchiectasis cohort, we performed a sensitivity analysis with different exclusion criteria using 2- and 3-year washout periods.”
page 7, lines 210–213
“Sensitivity analyses also revealed comparable HRs for incident aspergillosis in patients with bronchiectasis relative to those without bronchiectasis: 2-year (sdHR, 4.94; 95% CI, 3.16–7.71) and 3-year (sdHR, 4.94; 95% CI, 3.02–8.08) washout periods, respectively (Supplementary Table S1).”
C4. Figure 2: Figures 2B and 2C should be combined in the same figure, with the same applied for Figures 2D and 2E. This allows better visualisation of differences between the subgroups.
R4. Thank you for your comment. Reviewer 1 also recommended better visualisation of Figure 2. To respond to both reviewers’ recommendations, Figure 2A is now Figure 2 in the main manuscript, and we have combined Figures 2B, 2C, 2D, and 2E into one figure (Supplementary Figure S1) in the revised manuscript.

Reviewer 2 Report
This population-based retrospective cohort study evaluated the cumulative incidence and risk factors for aspergillosis in patients with non-CF bronchiectasis. The authors reported an increased incidence of aspergillosis at a HR of 4.53 in patients with non-CF bronchiectasis, compared to age- and sex-matched controls. The risk factors associated with the development of aspergillosis included comorbid pulmonary conditions (COPD, previous pulmonary tuberculosis, and NTM pulmonary disease).
Major comments
- Was there a minimum duration of follow-up for this cohort? If not, were patients who had follow-up duration < 1 year being excluded from this study?
- Why was diabetes mellitus not included in the Cox proportional hazards regression models?
- The statistical analyses for sensitivity analyses based on different criteria should be provided in the Methods section, with a summary of results presented in the Results section.
- Figure 2: Figures 2B and 2C should be combined in the same figure, with the same applied for Figures 2D and 2E. This allows better visualisation of differences between the subgroups.
Minor comments
There is a formatting error for Page 16, with the first paragraph of the Discussion being truncated into 2 pages.
Author Response

(The authors gave the same response as above.)

Round 2
Reviewer 2 Report
All comments have been adequately addressed.